# Highly Stable, Graphene-Wrapped, Petal-like, Gap-Enhanced Raman Tags

**DOI:** 10.3390/nano12101626

**Published:** 2022-05-10

**Authors:** Ming Chen, Bin Wang, Jingfan Wang, Hongliang Liu, Zhixiang Chen, Xiaoxuan Xu, Xing Zhao

**Affiliations:** 1Tianjin Key Laboratory of Micro-Scale Optical Information Science and Technology, Institute of Modern Optics, Nankai University, Tianjin 300350, China; dreamchenming@163.com (M.C.); wjf9816@163.com (J.W.); drliuhl@nankai.edu.cn (H.L.); dtchzx333@163.com (Z.C.); 2College of Artificial Intelligence, Nankai University, Tianjin 300350, China; wb@nankai.edu.cn

**Keywords:** gap-enhanced Raman tags, graphene, Raman-enhanced mechanism, stability

## Abstract

Gap-enhanced Raman tags (GERTs) were widely used in cell or biological tissue imaging due to their narrow spectral linewidth, weak photobleaching effect, and low biological matrix interference. Here, we reported a new kind of graphene-wrapped, petal-like, gap-enhanced Raman tags (GP-GERTs). The 4-Nitrobenzenethiol (4-NBT) Raman reporters were embedded in the petal-like nanogap, and graphene was wrapped on the surface of the petal-like, gap-enhanced Raman tags. Finite-difference time-domain (FDTD) simulations and Raman experimental studies jointly reveal the Raman enhancement mechanism of graphene. The *SERS* enhancement of GP-GERTs is jointly determined by the petal-like “interstitial hotspots” and electron transfer between graphene and 4-NBT molecules, and the total Raman enhancement factor (*EF*) can reach 10^10^. Mesoporous silica was grown on the surface of GP-GERTs by tetraethyl orthosilicate hydrolysis to obtain Raman tags of MS-GP-GERTs. Raman tag stability experiments showed that: MS-GP-GERTs not only can maintain the signal stability in aqueous solutions of different pH values (from 3 to 12) and simulated the physiological environment (up to 72 h), but it can also stably enhance the signal of different Raman molecules. These highly stable, high-signal-intensity nanotags show great potential for *SERS*-based bioimaging and multicolor imaging.

## 1. Introduction

Surface-enhanced Raman scattering (*SERS*) is an ultrasensitive vibrational spectroscopy technique that is widely used in various fields, such as chemistry, physics, biology, and medicine [1,2,3,4,5]. The nanoprobes designed based on *SERS* technology are called “*SERS* tags”, which are usually composed of metal nanoparticles and Raman reporter molecules [6]. The strong characteristic Raman signal can be generated by “*SERS* tags”; thus, it has a similar optical-labeling function with fluorescent dyes and quantum dots, showing great potential in the field of biological analysis and imaging [7,8,9]. However, this simple metal nanoparticle-Raman reporter molecule structure lacks stability, and the Raman signal is easily disturbed by the environment. Usually, a layer of material needs to be wrapped around the structure to protect the Raman signal molecule. When necessary, specific target molecules need to be modified outside the protective layer to obtain *SERS* probes with various biological functions.

Gap-enhanced Raman tags (GERTs) are an emerging class of *SERS* tags. In GERTs, Raman reporter molecules are embedded between metal core shells, which reduces the influence of the external environment and nanoparticle aggregation on Raman signals [10,11,12,13]. Recently, Zhang et al. [14]. proposed a petal-like, gap-enhanced Raman tag (P-GERTs), which enables single-particle detection due to the strong electromagnetic field hot spots generated by both the inner gap and the outer petaloid structure. However, the Raman reporters adsorbed on the surface of the outer petal structure of P-GERTs are still affected by the external environment, resulting in unstable *SERS* signals and poor controllability.

Graphene is one of the most widely studied and mature two-dimensional (2D) atomic materials [15,16,17]. It has the advantages of single atomic layer thickness, unique phononic structure, high-electron mobility, chemical inertness, and biocompatibility [18,19,20]. Moreover, graphene can also be used as a shell-separating nanoparticle extension material for the fabrication of ultrathin shells at the atomic layer scale, while also providing additional chemical enhancement [21]. Numerous studies have shown that the combination of graphene and metal nanoparticles can enhance the *SERS* performance of substrates [22,23,24]. Qiu et al. [25]. used the electrostatic interaction between graphene oxide and gold nanorods to synthesize graphene oxide-wrapped gold nanorods and reduce the biological toxicity of gold nanorods. Zhang’s research group [26,27] combined graphene with silver nanoparticles to obtain stable and highly reproducible *SERS* substrates. Li et al. [28]. reported the enhanced Raman spectra of graphene-wrapped gold nanoparticles, which exhibited good pH stability and high-temperature stability.

In this work, we developed new graphene-wrapped, petal-like, gap-enhanced Raman tags (GP-GERTs for brevity). The 4-NBT Raman reporters were embedded between the petal-like nano-gaps, and graphene was wrapped on the surface of the petal-like gold nanoparticles by electrostatic interaction. We explained the Raman enhancement mechanism of GP-GERTs through FDTD simulations and Raman experimental studies. The “interstitial hot spots” between petals and the “charge transfer” between graphene and 4-NBT jointly enhanced the *SERS* signal, and the total Raman *EF* can reach 1.75 × 10^10^. Mesoporous silica was grown on the surface of GP-GERTs by hydrolysis of tetraethyl orthosilicate to obtain MS-GP-GERTs. Due to the special structure of Raman tags and the chemical inertness of graphene, MS-GP-GERTs showed excellent stability in serum environment and aqueous solutions with different pH values. The Raman signal of MS-GP-GERTs remained stable, whether immersed in an aqueous solution with a pH ranging from 3 to 12 or immersed in a simulated physiological environment for a long time (up to 72 h). In addition, the MS-GP-GERTs structure can also achieve stable *SERS* enhancement for a variety of different Raman reporters. This highly stable, high-signal intensity Raman tag has great potential for applications in different types of biomedical imaging and multicolor Raman imaging.

## 2. Materials and Methods

### 2.1. Materials

Chloroauric acid (HAuCl_4_·4H_2_ O)) and cetyltrimethylammonium chloride (CTAC, 99%) were purchased from Shanghai Meryer Chemical Reagent Co., Ltd (Meryer, Shanghai, China). The 4-Nitrobenzenethiol (4-NBT, 95%), 1,4-benzenedithiol (1,4-BDT, 98%), 4-mercaptobenzonitrile (4-MBN, 95%), Biphenyl-4,4′-dithiol (B-4,4′-D, 98%), and 2-naphthalene thiophenol (2-NT, 98%) were obtained from Shanghai Macklin Biochemical Technology Co., Ltd (Macklin, Shanghai, China). Ascorbic acid (AR, 99%), sodium hydroxide solid powder (NaOH, AR, 96%), hydrochloric acid (ACS, 37%), sodium chloride (AR, 99.5%), glucose monohydrate (98%), and ethyl orthosilicate (TEOS) were purchased from Shanghai Aladdin Biochemical Technology Co., Ltd (Aladdin, Shanghai, China). Bovine serum albumin (98%) was purchased from Hefei Qiansheng Biotechnology Company (Qiansheng, Hefei, China). Mechanically exfoliated single-layer graphene (99%) was purchased from Shenzhen Suiheng Graphene Technology Co., Ltd (Suiheng, Shenzhen, China). Anhydrous ethanol and isopropanol were purchased from Tianjin Huaxun Pharmaceutical Technology Co., Ltd. (Huaxun, Tianjin, China). Deionized water (18.25 MΩ) was used for all experiments.

### 2.2. Synthesis of GP-GERTs and MS-GP-GERTs

Gold nanocores were synthesized by referring to the work of Zhang et al. [14]. The synthesis process of GP-GERTs and MS-GP-GERTs are shown in Figure 1.

Firstly, Raman reporter molecules were modified on the surface of gold nanocores. The 4-NBT ethanol solution (500 μL, 10 mM) was added to 10 mL of gold nucleus solution (1 nM) and sonicated for 30 min; the obtained sol particles were washed with CTAC solution (50 mM) by centrifugation and then re-dispersed in CTAC (50 mM) solution for later use. In this process, the ethanol solution of 4-NBT was replaced by the ethanol solution of different Raman molecules, and the gold nanocores modified with different Raman molecules can be obtained.

Secondly, gold nanopetals were grown on the surface of gold nanocores modified with Raman reporters. A total of 0.5 mL of 4-NBT molecule-modified gold nanonucleus solution was added to a mixed solution of CTAC (8 mL, 0.05 M), ascorbic acid (250 μL, 0.04 M) and chloroauric acid (500 μL, 5 mM). The solution changed from colorless to pink, purple, and blue, and it continued to be sonicated for 30 min. P-GERTs with 4-NBT Raman reports were obtained.

Thirdly, petal-gap enhanced Raman tags was wrapped by graphene. A total of 4 mL of the prepared P-GERTs were mixed with 2 mL of graphene ethanol solution (20 mg/L), and NaOH (0.05 M) was added to adjust the pH of the solution to 12.5. The mixture was stirred at 40 °C for 8 h; graphene could be bonded on the surface of P-GERTs via electrostatic interaction. After the dispersion was cooled to room temperature, it was centrifuged at 14,000 rcf for 15 min to remove the supernatant and was re-dispersed in CTAC to obtain GP-GERTs.

Finally, mesoporous silica shells were grown on the surface of GP-GERTs through TEOS hydrolysis. An amount of 1.5 mL of deionized water was added to 2 mL of GP-GERTs, and sodium hydroxide solution (0.05 M) was added dropwise to adjust the pH to 12.5. Next, 50 μL of 5% TEOS in isopropanol was added slowly to the solution under stirring. This procedure was repeated three times at 30-min intervals, and then the mixture was reacted for 24 h at 40 °C. The resulting solution was then centrifuged at 14,000 rcf for 15 min, the supernatant was removed, and the particles were redispersed in 2 mL of ethanol to obtain MS-GP-GERTs.

### 2.3. Characterization of GERTs

Scanning electron microscope (SEM) images were acquired on a Merlin Compact (Zeiss, Oberkochen, Germany) with an accelerating voltage of 10 kV. Transmission electron microscopy (TEM) images were acquired on a FEI Talos F200c (ThermoFisher, Waltham, MA, USA) at 120 kV. UV-Vis spectra were acquired using a Lambda 45 UV-Vis spectrophotometer (PerkinElmer, Waltham, MA, USA). Fourier transform infrared spectra (FTIR) were acquired using a VERTEX V70 Fourier infrared spectrometer (Bruker, Karlsruhe, Germany). Raman spectra were recorded with a laser confocal Raman spectrometer (XperRam200, Nanobase, Seoul, Korea) equipped with an Olympus upright microscope with a 40× objective, numerical aperture (NA) of 0.65, and an Andor scientific grade TE-cooled CCD. The Raman excitation wavelength was set to 785 nm, the power was 60 mW, the integration time was set to 1 s, and the spectral resolution was 2.5 cm^−1^. All Raman-testing experiments were performed on silicon substrates. The baseline of spectral data was removed by the LabSpec software (Horiba, Paris, France).

### 2.4. Stability Experiments of MS-GP-GERTs

For the experiment of PH stability, 2 mL of the prepared MS-GP-GERTs solution was centrifuged and dispersed in aqueous solutions of different pH values (pH values ranged from 3 to 12). After soaking for 30 min, it was centrifuged again and redispersed in 2 mL of ethanol. A small amount of the sample was drawn dropwise on the silicon substrate using a pipette, and the MS-GP-GERTs sample to be tested was obtained after natural drying.

Using 10% bovine serum albumin, 10% glucose solution, and 0.9% normal saline to simulate a physiological environment, the time stability of Raman tags in a physiological environment was tested. The specific test method is as follows: 2 mL of the prepared MS-GP-GERTs solution was centrifuged and dispersed in 2 mL of bovine serum albumin solution (10%). The mixing time of MS-GP-GERTs and 10% bovine serum albumin solution was controlled as 2 h, 12 h, 24 h, 36 h, and 72 h, respectively. It was then washed twice by centrifugation with absolute ethanol and redispersed in 2 mL of ethanol. A small amount of sample was drawn dropwise on the silicon substrate using a pipette, and the MS-GP-GERTs sample to be tested was obtained after natural drying. Two additional tests can be performed by replacing the 10% bovine serum albumin solution with 10% glucose solution or 0.9% normal saline.

## 3. Results

### 3.1. UV-Vis Absorption Spectroscopy

According to the synthesis process of Raman tags, we tested three sets of UV-Vis absorption spectroscopy for comparison. The red and black curves of Figure 2a represent the absorption spectra of gold nanocores and gold nanocores modified 4-NBT molecules, respectively. As shown in Figure 2a, the absorption peak of gold nanocores is around 524 nm, which is consistent with the absorption peak of 20 nm gold particles [29]. After the adsorption of 4-NBT molecules on the surface of the gold nanocores, the absorption peak has a red shift of 3 nm. The blue and green curves in Figure 2a represent the absorption spectra of P-GERTs and (MS)P-GERTs, respectively. Different from the absorption peak of gold nanocores, the absorption peak of P-GERTs is around 638 nm, which can be explained by the larger diameter of Raman tags and the generation of a large number of electromagnetic hot spots. In addition, compared with P-GERTs, the absorption peak of (MS)P-GERTs has a 5 nm red shift, which indicates that mesoporous silica can grow on the surface of P-GERTs [30].

Figure 2b is the UV-Vis absorption spectroscopy of graphene, GP-GERTs, and MS-GP-GERTs. As shown in Figure 2b, in the range of 300–900 nm, the absorption curve of graphene shows a decreasing trend, which is the same as that reported in the previous literature [31]. In addition, after the Raman tags were combined with graphene, the absorption peak of the Raman tags in the visible light range disappeared, which was mainly affected by graphene. It was also confirmed that the Raman tags were encapsulated by graphene, and GP-GERTs and MS-GP-GERTs were successfully synthesized.

### 3.2. Fourier Transform Infrared Spectroscopy (FTIR)

Figure 3 shows the FTIR spectra of graphene, GP-GERTs, and MS-GP-GERTs. As shown in Figure 3, graphene has no obvious characteristic peaks in the infrared region, which is because the single-layer graphene we use is prepared by mechanical exfoliation. During the measurement of FTIR spectra, graphene is compressed into graphite flakes. Therefore, none of the peaks were measured. GP-GERTs also showed the same trend, with no obvious peaks in the infrared region, which also indicated that graphene was wrapped on the surface of petal-like gold nanostructures. The FTIR spectra of MS-GP-GERTs showed characteristic peaks of mesoporous silica. The peak around 790 cm^−1^ is derived from the stretching vibration of Si-O bond, and the peak around 1075 cm^−1^ is derived from the Si-O-Si group [32]. Furthermore, since our mesoporous silica was obtained by hydrolysis of TEOS (isopropanol as solvent), and the composite structure was not calcined when measuring the FTIR spectrum. Therefore, the vibration peaks of C-H bond appeared at 2850 cm^−1^ and 2920 cm^−1^, and the vibration peak of the Si-OH bond appeared near 3014 cm^−1^ [33].

### 3.3. SEM and TEM

Figure 4 shows the SEM images of P-GERTs. Figure 4a is a 100 KX magnification of P-GERTs, and the scale bar is 100 nm. The shape of P-GERTs nanoparticles is similar to that of petals, and the agglomeration between particles is not obvious, and they are relatively independent of each other. Figure 4b is the P-GERTs magnified by 200 KX, and the scale bar is 20 nm. According to Figure 4b, the morphology of P-GERTs nanoparticles can be better observed, and the diameter of the particles can be estimated. The SEM images confirmed that the P-GERTs had a petal-like shape with particle diameters ranging from about 60 nm to 80 nm.

Figure 5a shows the TEM image of P-GERTs, and the scale bar is 20 nm. The microstructure of P-GERTs can be more clearly observed from Figure 5a, which consists of an inner gold core, middle nano gaps (marked by a red arrow in Figure 5a), and outer nano petals. The total diameter of P-GERTs is about 70 nm, which is mutually confirmed with the previous SEM images. Figure 5b,c are TEM images of monolayer graphene and GP-GERTs, respectively, and the scale bar is 2 μm. The complete morphology of the monolayer graphene can be seen from Figure 5b, and combined with Figure 5c, it can be clearly seen that most of the P-GERTs nanoparticles are covered by the mesh-like monolayer graphene. Figure 5d shows a TEM image of (MS)P-GERTs with a scale bar of 200 nm, and the upper right inset shows a further magnified single (MS)P-GERTs particle with a scale bar of 20 nm. As shown in Figure 5d, the mesoporous silica layer wraps around the P-GERTs with a thickness of about 15–25 nm, which also shows a clear mesoporous structure in the enlarged inset. Figure 5e shows the TEM picture of MS-GP-GERTs, and the scale bar is 200 nm. From Figure 5e, it can be seen that most of the MS-GP-GERTs nanoparticles are dispersed independently in the solution, and the individual nanoparticles are marked by red arrows, and the diameter of the particles is about 80–95 nm. It can be observed that a mesoporous silica layer grows on the surface of P-GERTs while being covered by excess graphene in solution. It should be pointed out that there is also a very thin graphene layer (about 0.5–1 nm) between the mesoporous silica layer and the P-GERTs particle. This is because the excess monolayer graphene was well mixed with the P-GERTs before the mesoporous silica cladding is grown. Unfortunately, such thin graphene is difficult to observe together with mesoporous silica and P-GERTs in TEM characterization. However, in our later Raman experiments and calculations of the enhancement mechanism, the graphene interlayer was proven to exist.

### 3.4. Simulation

FDTD solutions (Lumerical, Vancouver, BC, Canada) simulation software was used to simulate the spatial distribution of the electromagnetic field strength of the Raman tags. Figure 6a shows a simulation model for simulating the electric field effect of a Raman tag. As shown in Figure 6a, the laser source is total-field scattered-field, and the laser is polarized along the X direction and propagates in the -Z direction. The real structures of P-GERTs are very complex with random petal-like structures, and in the FDTD simulations, we used small gold nanospheres to simulate the petal structures. Figure 6c shows the 3D model of the P-GERTs we used, which consists of an inner gold core (radius 15 nm), a molecular layer of 4-NBT in the middle (thickness 1 nm), and outer petals (small gold spheres with a radius of 9 nm). A total of 26 small golden spheres were used to simulate the petals of P-GERTs. A total of 18 of them were evenly distributed on three circles perpendicular to the X, Y, and Z axes, and the distance between the center of the small gold sphere and the center of the inner gold core is 25 nm. The remaining eight were distributed in the center of the remaining blank area, and the distance between the centers of the spheres is also 25 nm. The total particle size of the P-GERTs is about 68 nm, and the specific size is given in Figure 6b. As shown in Figure 6d–f, the frequency domain filed and power monitor are placed in the XY plane, and the 4-NBT molecular layer refractive index was set to one, according to other literatures [34,35]. The graphene layer thickness was set to 1 nm, and the refractive index of graphene was set to 2.63 + 1.28 i [36]. The thickness of the SiO_2_ layer was set to 20 nm, and the refractive index of SiO_2_ and Au were both derived from the parameters in the software. The calculation priority is from inside to outside, and the refractive index of the medium surrounding the Raman label was set to 1.33 (simulated water environment).

To investigate the effect of different laser wavelengths on the electric field enhancement effect of P-GERTs, we calculated the electromagnetic field strengths of P-GERTs when excited at three different laser wavelengths (532, 638, and 785 nm). As shown in Figure 7a–c, the electric field intensity distributions of single P-GERTs were similar under the excitation of three laser wavelengths, and the electromagnetic hot spots were mainly concentrated in the gap between the gold core and the petals. When excited by 638 nm laser, the maximum ratio of electromagnetic field strength is 38.6, when excited by 785 nm laser, the maximum ratio is 33, and when excited by 532 nm laser, the maximum ratio is 28.7. According to the previous absorption spectrum of P-GERTs, the resonance absorption peak of P-GERTs is around 638 nm; thus, 638 nm belongs to resonance excitation, which is also the reason for the strongest electric field enhancement effect. However, resonance excitation can lead to a significant endothermic effect of metal nanoparticles, and the temperature around the nanoparticles increases rapidly, which in turn affects the stability of Raman tags [37]. Both 785 nm and 532 nm belong to non-resonant excitation, and the thermal effect is not obvious, which can ensure the long-term stability of the Raman tags. Compared with 532 nm, the electric field enhancement effect is stronger at 785 nm excitation; thus, the excitation light of 785 nm wavelength is used in the subsequent simulation and Raman experiments.

Figure 7d–f shows the electric field enhancement distributions for single (MS)P-GERTs, GP-GERTs, and MS-GP-GERTs. As shown in Figure 7d–f, the electromagnetic hot spots of single (MS)P-GERTs, GP-GERTs, and MS-GP-GERTs are also concentrated in the gap between the gold core and the petals. Silica layer and graphene layer have little effect on the electromagnetic field strength of P-GERTs. The maximum ratio of electromagnetic field strength of single (MS)P-GERTs, GP-GERTs, and MS-GP-GERTs were 32.3, 33.8, and 33.5, respectively. Several studies have shown that graphene has the effect of enhancing the Raman signal, but the enhancement mechanism is chemical enhancement, and the enhancement effect of graphene on the electromagnetic field is not obvious [38,39,40]. Our simulation results also provide support for this claim.

### 3.5. Raman Spectra and Enhancement Mechanism of GERTs

Figure 8a shows the Raman spectrum of 4-NBT molecule (10^−3^ mol/L) on silicon substrate. As shown in Figure 8a, there are five main characteristic peaks in the Raman spectrum of 4-NBT molecule, which are 1335 cm^−1^, 722 cm^−1^, 854 cm^−1^, 1080 cm^−1,^ and 1570 cm^−1^, respectively. The peak of 1335 cm^−1^ corresponds to the strong mode ν(NO_2_), the remaining four peaks correspond to the four weak modes, π(CH) + π(CS) + π(CC), π(CH), ν(CS), and ν(CC), respectively. Figure 8b shows the Raman spectrum of graphene (10^−3^ mol/L) on silicon substrate. Within 1200–2800 cm^−1^, there are three Raman characteristic peaks of graphene, which are D peak at 1330 cm^−1^, G peak at 1575 cm^−1^, and 2D peak at 2670 cm^−1^. Figure 8c shows the Raman spectra of four Raman tags of P-GERTs, (MS)P-GERTs, GP-GERTs, and MS-GP-GERTs. As shown in Figure 8c, the Raman peaks of all Raman tags are similar to those of the 4-NBT molecule. We found that the Raman signal intensities of P-GERTs and (MS)P-GERTs were comparable, and the Raman signal intensities of GP-GERTs and MS-GP-GERTs were comparable. We can clearly see that after graphene-wrapped P-GERTs, the Raman signal of GP-GERTs is significantly enhanced. The same conclusion can be drawn by comparing the Raman signal intensities of MS-GP-GERTs and (MS)P-GERTs. To determine the concentration detection limit of MS-GP-GERTs Raman tags, the prepared MS-GP-GERTs Raman tag solutions were diluted with ethanol. We prepared samples with Raman tags of MS-GP-GERTs at concentrations of 100 pM, 10 pM, 1 pM, 100 fM, and 10 fM for Raman measurement, and the experimental results obtained are shown in Figure 8e. From Figure 8e, it can be seen that even at a concentration of 100 fM of MS-GP-GERTs, a 4-NBT Raman curve with a good signal-to-noise ratio can be obtained. At further dilution to 10 fM, the Raman signal of 4-NBT was hardly detected.

It is generally reported that there are two enhancement mechanisms of *SERS* effect, namely physical enhancement mechanism and chemical enhancement mechanism [41,42,43]. The physical enhancement mechanism can be explained as the “interstitial hot spots” of rough metal surfaces or nanostructures are excited by the laser, causing localized surface plasmon resonance to generate electromagnetic field enhancement (EM). The chemical enhancement mechanism can be regarded as the signal enhancement caused by the charge transfer between the substrate and the molecule, namely the charge transfer mechanism (CT). It is worth mentioning that the graphene D peak (1330 cm^−1^) used in our experiments is very close to the strong mode ν(NO_2_) peak (1335 cm^−1^) of the 4-NBT molecule, and the graphene G peak (1575 cm^−1^) is very close to the ν(CC) peak (1570 cm^−1^) of the 4-NBT molecule. Coincidentally, in our experiments, the most obvious Raman signal enhancements are also the two peaks at 1335 cm^−1^ (the yellow-shaded area in Figure 8c) and 1570 cm^−1^ (the red-shaded area in Figure 8c). According to Zhu et al. [44], the intensity ratio of the D peak and the G peak in the Raman spectra will change during the process of converting graphene to graphene oxide. Due to the increased defects in graphene oxide, the D peak will eventually become stronger. To determine the source of the Raman peak enhancement in Figure 8c, we replaced the 4-NBT ethanol solution with ethanol solution during the synthesis of MS-GP-GERTs and GP-GERTs so that the final composite structure does not contain 4-NBT molecules. Figure 8d shows the Raman spectra of the composite structures, MS-GP-GERTs, GP-GERTs, and graphene. As shown in Figure 8d, although the signal intensities of graphene D peak (1330 cm^−1^) and G peak (1575 cm^−1^) in MS-GP-GERTs and GP-GERTs were slightly enhanced, the ratio of the two hardly changed. This indicates that the graphene in the composite structure has not changed. It was further confirmed that the enhancement of Raman peak in Figure 8c originated from the *SERS* signal of 4-NBT molecules.

Therefore, the Raman signal enhancement mechanism of GP-GERTs tags may have three situations: one can be explained as CT mechanism, the electron transfer between graphene and 4-NBT molecules causes the Raman signal enhancement of GP-GERTs; the other one can be explained as the EM mechanism, the “interstitial hotspot” around the P-GERTs tag enhances the Raman signal of graphene, leading to the enhancement of the Raman signal of GP-GERTs; the third possibility is that these two enhancement mechanisms coexist.

According to the principle of electromagnetic field enhancement, the approximate electromagnetic enhancement factor (*EF_EM_*) can usually be calculated with the following formula [45]:(1)$$EF_{EM}=\frac{{\left|E_{out}(\omega_{0})\right|}^{2}{\left|E_{out}(\omega_{s})\right|}^{2}}{{\left|E_{0}\right|}^{4}}\approx \frac{{\left|E_{out}(\omega_{0})\right|}^{4}}{{\left|E_{0}\right|}^{4}}$$
where $E_{0}$ is the incident electric field strength, which is usually set to 1 V/m. $E_{out}\left(\omega_{0}\right)$ is the local electric field strength of incident light (frequency $\omega_{0}$), and $E_{out}\left(\omega_{s}\right)$ is the local electric field strength of Raman scattered light (frequency $\omega_{s}$).

According to the electric field simulation results in Section 3.3, the *EF_EM_* of P-GERTs, (MS)P-GERTs, GP-GERTs, and MS-GP-GERTs Raman tags can be calculated to be 1.18 × 10^6^, 1.09 × 10^6^, 1.31 × 10^6^, and 1.26 × 10^6^, respectively.

On the other hand, in the actual Raman experiment, the following formula can be used to calculate the enhancement factor (*EF*) of the experimental results [46]:(2)$$EF=\frac{I_{SERS}/N_{SERS}}{I_{R\mathrm{aman}}}/N_{R\mathrm{aman}}$$

Among them, EF represents the enhancement factor obtained by analyzing the experimental results. $I_{SERS}$ and $I_{R\mathrm{aman}}$ represent the Raman intensity of *SERS* and the Raman intensity of the molecule itself, respectively. $N_{SERS}$ and $N_{R\mathrm{aman}}$ represent the number of molecules in the *SERS* experiment and the number of molecules in the ordinary Raman experiment, respectively.

For normal Raman measurements, the concentration of the 4-NBT Raman molecule is 1 mM. For the calculation of the number of molecules in the *SERS* experiment, according to literature, the surface area of gold nanospheres with a diameter of 20 nm is about 1256 nm^2^, and it can be assumed that the adsorption area of each Raman molecule is 0.2 nm^2^ [47,48]. We estimate the number of Raman molecules adsorbed on each P-GERT to be 6280, and multiplying the concentration of gold nanocores can calculate the number of 4-NBT Raman molecules in the *SERS* experiment. The enhancement factors of P-GERTs, (MS)P-GERTs, GP-GERTs, and MS-GP-GERTs can be calculated according to the signal intensity of 4-NBT molecule at 1335 cm^−1^ in the Raman experiment and Equation (2), respectively, $EF=7.56\times 10^{9}$, $EF=7.95\times 10^{9}$, $EF=1.67\times 10^{10}$, and $EF=1.75\times 10^{10}$. According to the signal intensity of graphene at 1575 cm^−1^ in the Raman experiment and Equation (2), the enhancement factors of GP-GERTs and MS-GP-GERTs can be calculated to be $EF=5.3\times 10^{9}$ and $EF=4.34\times 10^{9}$, respectively. Compared with the previous calculation results of the electromagnetic enhancement factor (*EF_EM_*), *EF_EM_* is 10^6^ order of magnitude, while the actual *EF* in the experiment is 10^9^~10^10^ order of magnitude, and the difference between *EF_EM_* and actual *EF* is 10^3^~10^4^ order of magnitude. This indicates that the Raman signal enhancement of GP-GERTs comes not only from EM enhancement, but also from chemical enhancement (CT mechanism), and the Raman signal enhanced by CT mechanism is in the order of 10^3^~10^4^.

Based on the experimental and theoretical calculation results, we propose a schematic diagram of the possible Raman enhancement mechanism of GP-GERTs tags. As shown in Figure 9, the EM mechanism and the CT mechanism work together on 4-NBT molecules and graphene. The EM mechanism originates from “electromagnetic hot spots” inside and on the surface of the petal-like nanotags. The CT mechanism mainly comes from the following three aspects: Firstly, the gold nanoparticles are excited by light to generate hot electrons, which are transferred to 4-NBT molecules, and the 4-NBT molecules are excited to generate *SERS* signals. Secondly, the hot electrons on the gold nanoparticles may also be transferred to the graphene to enhance the Raman signal of the graphene or transferred to the 4-NBT molecule after passing through the graphene. Thirdly, the hot electrons generated by the photo-excited graphene are transferred to the 4-NBT molecule, which in turn generates the *SERS* signal.

### 3.6. Stability of MS-GP-GERTs

The stability of the Raman signal is an important indicator for evaluating Raman tags. A stable Raman signal for a long time is an important basis for the practical application value of Raman tags. For Raman tags applied in the field of bioimaging, it is usually necessary to maintain stability under various storage conditions and physiological environments (such as different pH and serum solutions). Figure 10a shows the experimental data of the stability study of MS-GP-GERTs in aqueous solutions with different pH values. As shown in Figure 10a, the Raman spectra and signal intensities of MS-GP-GERTs were hardly affected despite the wide range of pH changes (from pH = 3 to pH = 12). This indicates that MS-GP-GERTs have strong pH stability, especially in acidic environments, and the Raman tags reported previously are difficult to maintain in acidic solutions (pH < 5) [14,30] since the Raman 2D peak of graphene is more easily affected by the pH of the solution [49,50]. To evaluate the stability of graphene quality, similar to Figure 8d, we investigated the changes of the 2D peaks of graphene in the composite structure MS-GP-GERTs without 4-NBT molecules in aqueous solutions with different pH values. The experimental results are shown in Figure 10b. It can be seen from Figure 10b that the 2D peaks of graphene were less affected by the pH values of the solution, and the intensity of the 2D peaks hardly changes. Only when the pH value was 3, the 2D characteristic peak was shifted slightly to lower wave numbers. Therefore, we believe that the quality of graphene in the composite structure MS-GP-GERTs is stable in solutions of different pH values.

Subsequently, we carried out a temporal stability test of the Raman signal of MS-GP-GERTs in 10% bovine serum albumin solution. As shown in Figure 10c, MS-GP-GERTs were stable for a long time in 10% bovine serum albumin solution. After 72 h, the Raman spectral curves and signal intensities of MS-GP-GERTs remained basically unchanged, showing excellent temporal stability in a simulated serum physiological environment. Considering the actual physiological environment, we further investigated the stability of MS-GP-GERTs in glucose solution and normal saline. The concentration of glucose solution was set to 10%, and 0.9% normal saline was prepared with sodium chloride and deionized water. Similar to 10% bovine serum albumin, the incubation time of MS-GP-GERTs with glucose solution (10%) and normal saline (0.9%) was varied from 0 h to 72 h, and the obtained experimental results are shown in Figure 10d,e. From Figure 10d, it can be seen that the Raman spectra of MS-GP-GERTs are very stable during the incubation time, and the Raman tags show excellent temporal stability in glucose solution. Figure 10e shows the change of the normalized Raman intensity of the 1335 cm^−1^ Raman peak during the incubation of MS-GP-GERTs with normal saline. The Raman intensity of MS-GP-GERTs was also stable during the incubation time, and saline had little effect on the stability of the Raman signal.

For most *SERS* tags, due to the direct adsorption of Raman reporters on the surface of metal substrates, it is difficult to maintain stable performance under harsh conditions, such as long-term laser irradiation and strong acid conditions [51,52,53]. This limits the application of *SERS* tags to a certain extent, especially in in vivo imaging applications. In our experiments, MS-GP-GERTs showed excellent stability in the serum environment and aqueous solutions with different pH values. The main reasons are analyzed as follows: First, our Raman reporters are not simply adsorbed on the surface of the metal substrate but embedded in the inner gap of the petal-shaped gold nanoparticles, which makes the Raman reporters less disturbed by the external environment. Secondly, graphene is wrapped on the surface of P-GERTs, and the chemical inertness of graphene can protect Raman reporters from harsh environments. In addition, the protective layers of the mesoporous silica grow on the surface of the graphene-wrapped, petal-like tags, which further improve the stability and biocompatibility of the Raman tags.

Finally, we changed the Raman reporters inside the MS-GP-GERTs to study the stability of the graphene-wrapped, petal-like, gap-enhanced Raman tag structure to the *SERS* enhancement of different Raman reporters. Figure 11a shows the schematic structure of MS-GP-GERTs with different Raman reporters. Different colors correspond to different Raman reporters, and red, green, blue, and purple correspond to 1,4-BDT molecules, 4-MBN molecules, B-4,4′-D, and 2-NT molecules, respectively. Figure 11b shows the *SERS* spectra corresponding to Figure 11a. As can be seen in Figure 11b, the graphene-wrapped, petal-like, gap-enhanced Raman tag structure can achieve stable *SERS* enhancement for all four Raman reporter molecules. Similar to Figure 8e, we further examined the concentration detection limit of MS-GP-GERTs tags with different Raman reporters. Figure 11c,d show the Raman signals of MS-GP-GERTs tags with different Raman reporters at concentrations of 10 pM and 100 fM, respectively. Raman spectra of different reporter molecules with good signal-to-noise ratio can also be obtained when the Raman tags concentration is 100 fM. This means that *SERS* tags with different Raman characteristic peaks can be easily prepared by only replacing the internal Raman reporters of MS-GP-GERTs. It shows the great potential of MS-GP-GERTs in different types of biomedical imaging and multicolor Raman imaging.

## 4. Conclusions

In summary, we developed new graphene-wrapped, petal-like, gap-enhanced Raman tags and demonstrated the detailed synthesis process. We explained the specific role of graphene in Raman tags through FDTD simulations and Raman spectroscopy experimental studies. The Raman enhancement of GP-GERTs is determined by both CT mechanism and EM mechanism, and the total Raman *EF* can reach 10^10^. The stability experiments show that MS-GP-GERTs not only have excellent stability in different PH values (from 3 to 12) aqueous solutions, but also maintain signal stability in the simulated physiological environment for a long time (up to 72 h). In addition, the MS-GP-GERTs structure can also achieve stable enhancement of different Raman reporters, showing a bright application prospect in the field of biomedical imaging.

## Figures and Tables

**Figure 1 nanomaterials-12-01626-f001:**
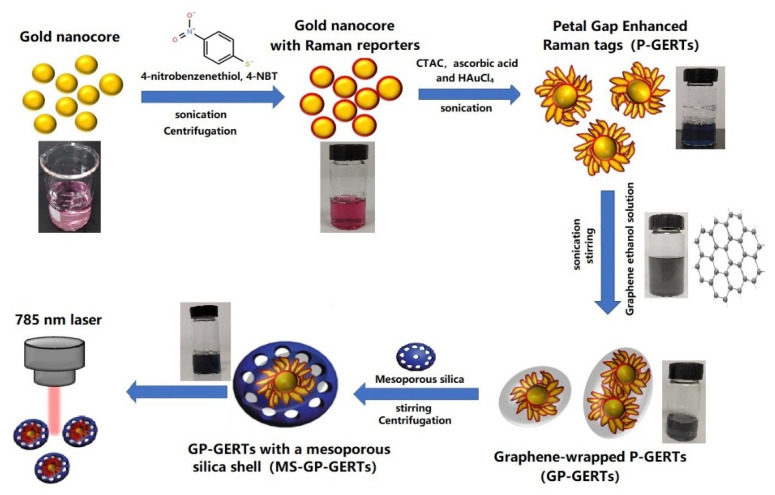
Schematic illustration of the synthesis process of the GP-GERTs and MS-GP-GERTs and *SERS* measurement by Raman system.

**Figure 2 nanomaterials-12-01626-f002:**
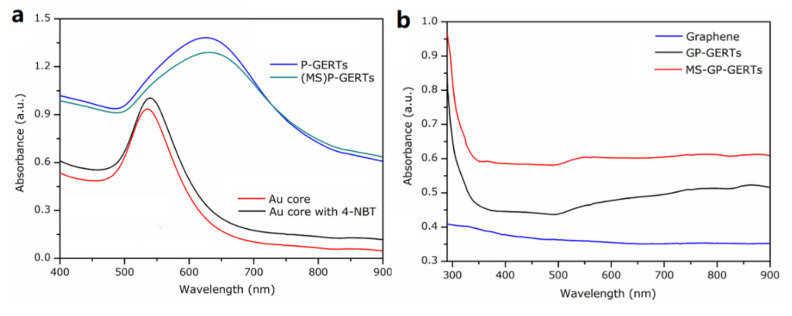
UV-Vis Absorption Spectroscopy: (**a**) absorption spectroscopy of gold core, gold core modified Raman molecule, P-GERTs, and (MS) P-GERTs; (**b**) UV-Vis absorption spectroscopy of graphene, GP-GERTs and MS-GP-GERTs.

**Figure 3 nanomaterials-12-01626-f003:**
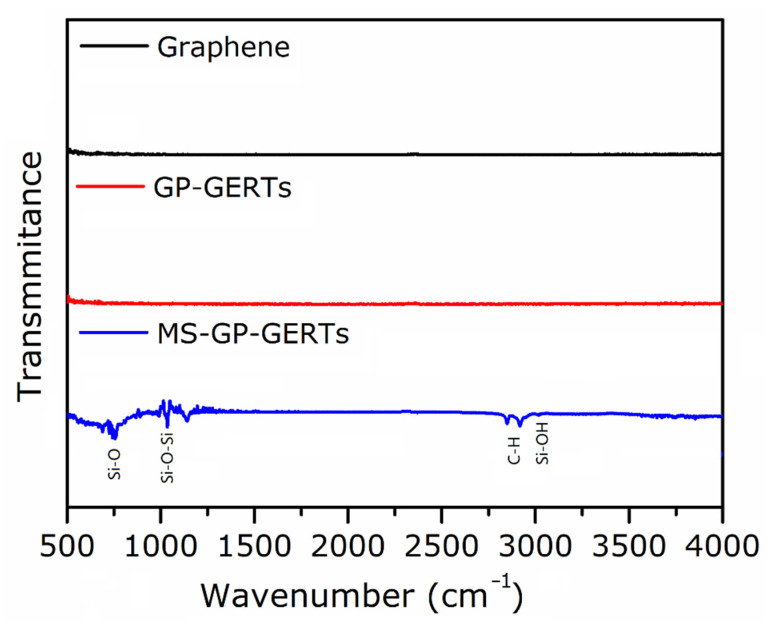
FTIR spectra of graphene, GP-GERTs, and MS-GP-GERTs.

**Figure 4 nanomaterials-12-01626-f004:**
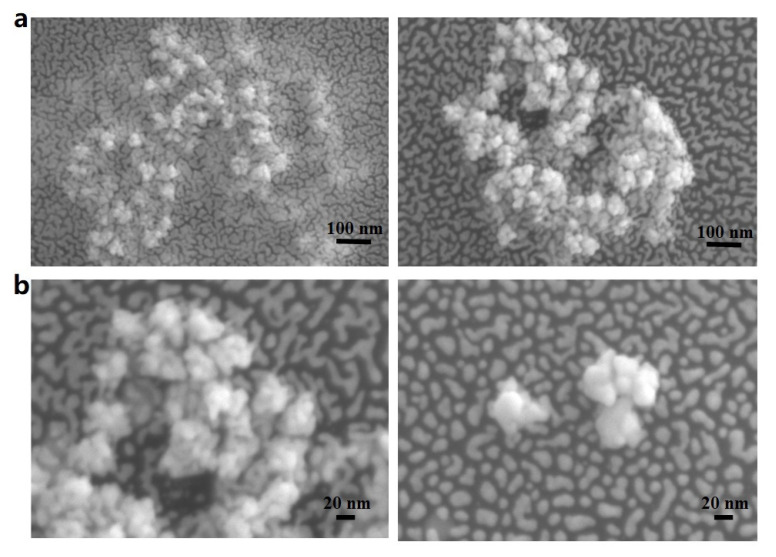
SEM images of P-GERTs: (**a**) 100 KX magnification; (**b**) 200 KX magnification.

**Figure 5 nanomaterials-12-01626-f005:**
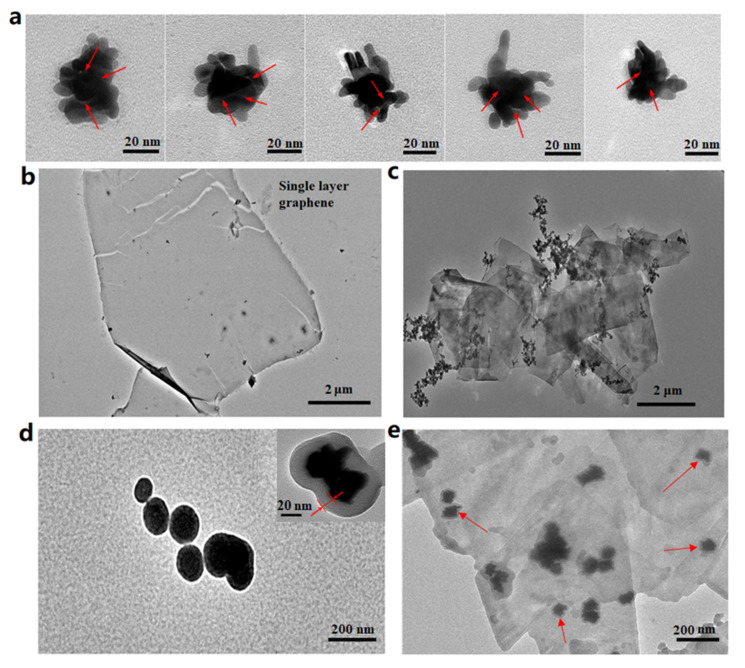
(**a**) TEM image of P-GERTs; (**b**) TEM image of monolayer graphene; (**c**) TEM image of GP-GERTs; (**d**) TEM image of (MS) P-GERTs; (**e**) TEM image of MS-GP-GERTs.

**Figure 6 nanomaterials-12-01626-f006:**
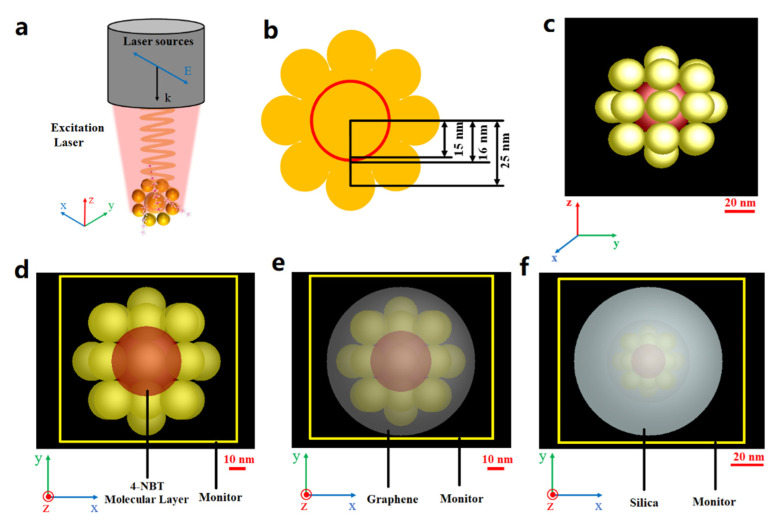
(**a**) Schematic diagram of the Raman tag electric field effects simulation model; (**b**) 2D dimension drawing of P-GERTs simulation model; (**c**) 3D simulation model of P-GERTs, the scale bar is 20 nm; (**d**–**f**) Monitor setup planes for P-GERTs, GP-GERTs, and MS-GP-GERTs, the scale bars are 10 nm, 10 nm and 20 nm, respectively.

**Figure 7 nanomaterials-12-01626-f007:**
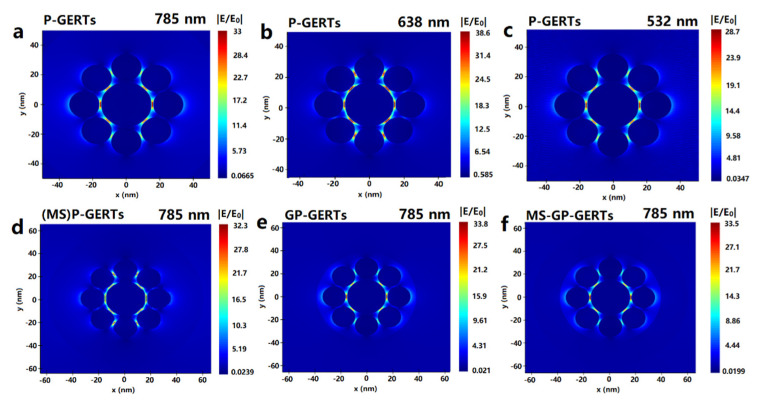
(**a**–**c**) Electric field enhancement distributions of single P-GERTs calculated by FDTD at excitation wavelengths of 785, 638, and 532 nm; (**d**–**f**) Electric field enhancement distributions of single (MS)P-GERTs, GP-GERTs, and MS-GP-GERTs, all with excitation wavelengths of 785 nm.

**Figure 8 nanomaterials-12-01626-f008:**
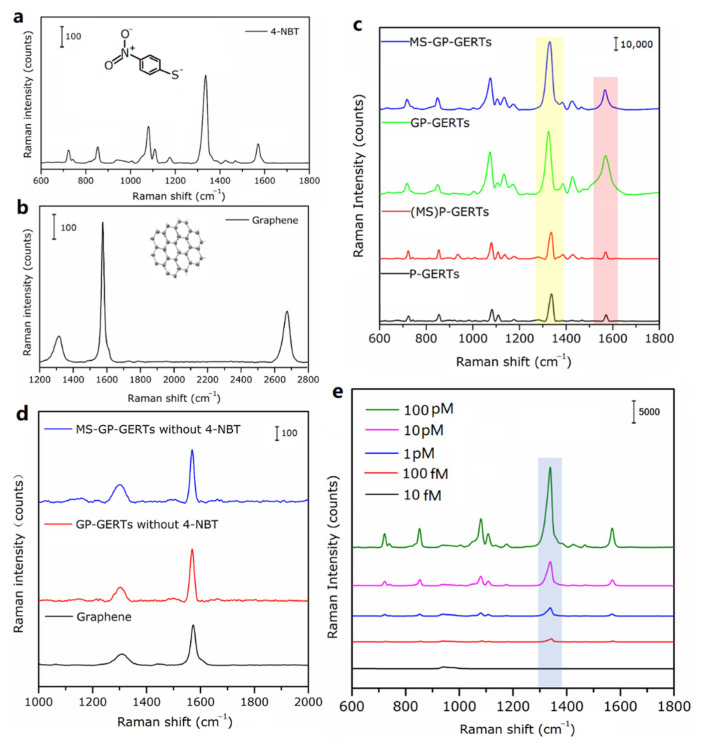
(**a**) Raman spectrum of 4-NBT on silicon substrate; (**b**) Raman spectrum of graphene on silicon substrate; (**c**) Raman spectra of P-GERTs, (MS)P-GERTs, GP-GERTs, and MS-GP-GERTs; (**d**) Raman spectra of composite structures MS-GP-GERT, GP-GERT, and graphene without 4-NBT molecules; (**e**) Raman spectra of different MS-GP-GERTs Raman tag concentrations.

**Figure 9 nanomaterials-12-01626-f009:**
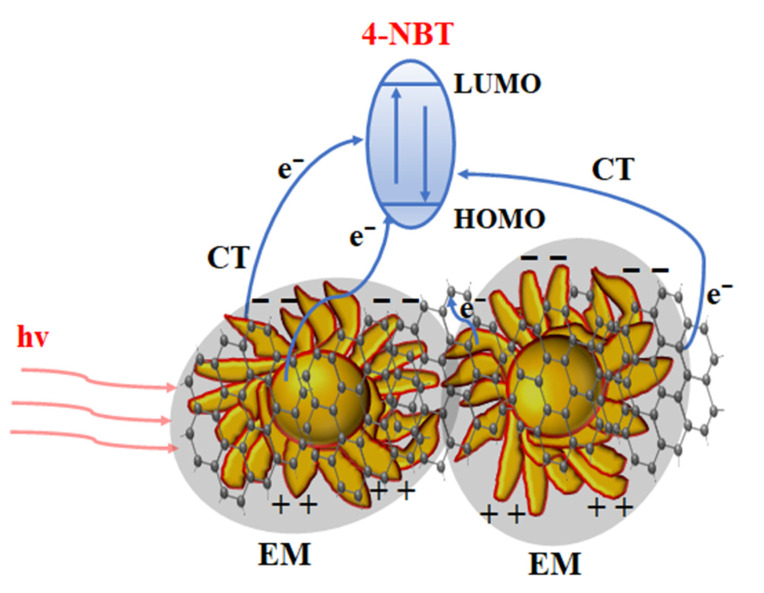
Schematic of Raman enhancement mechanism of GP-GERTs.

**Figure 10 nanomaterials-12-01626-f010:**
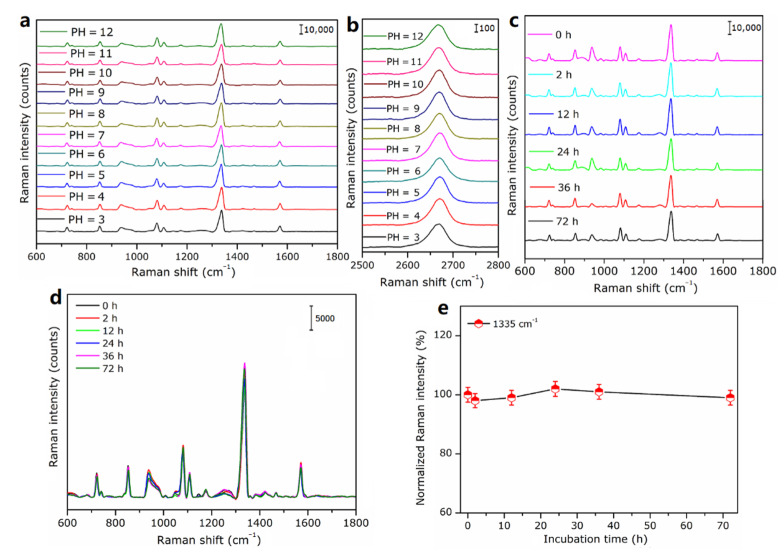
(**a**) *SERS* spectra of MS-GP-GERTs in aqueous solutions with different pH values; (**b**) 2D peak Raman curves of graphene in composite MS-GP-GERTs in aqueous solutions with different pH values; (**c**) *SERS* spectra of MS-GP-GERTs in 10% bovine serum albumin solution at different incubation times; (**d**) *SERS* spectra of MS-GP-GERTs in 10% glucose solution at different incubation times; (**e**) variation curve of normalized Raman intensity at 1335 cm^−1^ during incubation of MS-GP-GERTs with normal saline.

**Figure 11 nanomaterials-12-01626-f011:**
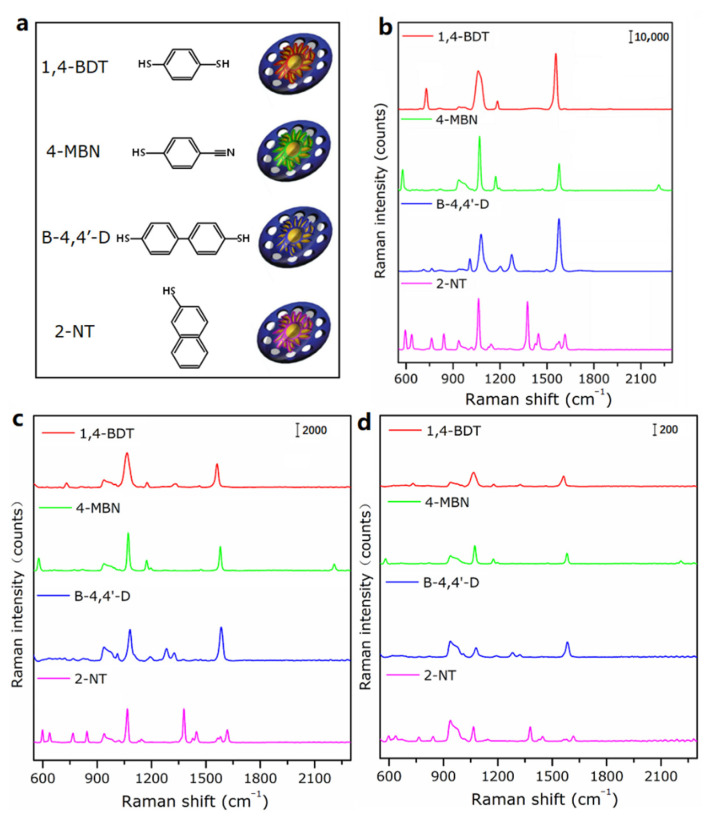
(**a**) Schematics; (**b**) Raman spectra of MS-GP-GERTs with different Raman reporters of 1,4-BDT (red), 4-MBN (green), B-4,4′-D (blue), and 2-NT (purple) immobilized in the nanogaps; (**c**) Raman spectra of MS-GP-GERTs (10 pM) with different Raman reporters; (**d**) Raman spectra of MS-GP-GERTs (100 fM) with different Raman reporters.

## Data Availability

The data presented in this article are available on request from the corresponding author.

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
