# Peer review of "Highly Stable, Graphene-Wrapped, Petal-like, Gap-Enhanced Raman Tags"

_nanomaterials, 2022, doi:10.3390/nano12101626_

Round 1

Reviewer 1 Report

General comments:

The authors fabricated gap-enhanced Raman tags (GERTs) composed of dyes sandwiched with gold nanoparticles, graphene, and mesoporous silica. They investigated the surface enhanced Raman scattering (SERS) enhancement mechanism of the GERTs by experiment and finite-difference time-domain (FDTD) numerical calculation.

However, the results are not enough to lead to a conclusion. Also, there are several technical problems in the manuscript.

Therefore, I think that minor revision is needed to bring this study to the level required for publication in Nanomaterials.

Specific Comments:

[1] The authors should use abbreviations correctly. They should uniquely define the abbreviations and for the first time that the term is used within the manuscript and then used throughout the remainder of the manuscript. For example, the term “gap-enhanced Raman tags” is used several times after the definition of abbreviation. Additionally, “FDTD” and “EF” are not defined.

[2] In Figure 5c-5f, a scale bar should be added. Also, the aspect ratio of panel 5c should be fixed as 1. Additionally, the direction of the x-y axes should be unified between Figures 5 and 6.

[3] The authors describe the unit of electric field enhancement as V/m. However, in my understanding, the electric field enhancement in FDTD calculation is dimensionless because this value is normalized by E0, as shown in Figure 6.

[4] As the authors describe, the electromagnetic (EM) enhancement factor of SERS is calculated from both incident and scattering electric field strengths. I wonder how the authors obtain the scattering electric field with the Stokes Raman shift. I recommend that the authors calculate the near-field spectra, which are an electric field enhancement as a function of wavelength, to show the incident and scattering electric field enhancements clearly.

[5] In line 337, the authors claim that the Raman signal enhancement on GP-GERT is derived from the charge transfer (CT) mechanism in addition to the EM mechanism. They also mention that the CT comes from hot-electron transfer from Au or graphene to dye molecule. However, they do not show direct evidence to prove the CT mechanism, such as absorption spectral change and theoretical calculation.

Reviewer 2 Report

In this study, the authors reported on the preparation of complex graphene-wrapped tags for Raman signal improvement. In general, the article can be considered for publication after a revision process.  

1) Abstract. All abbreviations should be defined first and later used in the text, check for example FDTD, EF, 4-NBT, etc. The manuscript should be polished by a native speaker. 
2) The authors need to perform the structural analysis of as-prepared samples, in particular XRD and FTIR. 
3)The final composite should be also tested by XPS and/or Raman, in particular, it is important to know whether the final form is graphene, reduced graphene oxide, or graphene oxide. 
4) Figure 7 - please confirm that the Raman signal of 4-NBT at different concentrations also shows improved peaks as compared to other tags. A similar approach can be also used for data in Figure 10.  
5) Figure 9. Other interferent molecules can be introduced to check whether the Raman signal is still stable.   

Round 2

Reviewer 2 Report

A revised manuscript can be accepted for publication. 

Author Response

Thank you for your outstanding work. We appreciate you raising valuable advices to help us to improve the manuscript.